# RepoCoder: Repository-Level Code Completion Through Iterative Retrieval and Generation

**Fengji Zhang[1], Bei Chen[2], Yue Zhang[2], Jacky Keung[1], Jin Liu[3],**
**Daoguang Zan[2], Yi Mao[2], Jian-Guang Lou[2], Weizhu Chen[2]**
[1]City University of Hong Kong, [2]Microsoft Corporation, [3]Wuhan University
fengji.zhang@my.cityu.edu.hk, jacky.keung@cityu.edu.hk, jinliu@whu.edu.cn
{beichen, zhayue, v-dazan, maoyi, jlou, wzchen}@microsoft.com

## Abstract

The task of repository-level code completion is to continue writing the unfinished code based on a broader context of the repository. While for automated code completion tools, it is difficult to utilize the useful information scattered in different files. We propose RepoCoder, a simple, generic, and effective framework to address the challenge. It streamlines the repository-level code completion process by incorporating a similarity-based retriever and a pre-trained code language model in an iterative retrieval-generation pipeline. RepoCoder makes effective utilization of repository-level information for code completion and has the ability to generate code at various levels of granularity. Moreover, we propose a new benchmark RepoEval, which consists of the latest and high-quality real-world repositories covering line, API invocation, and function body completion scenarios. Experimental results indicate that RepoCoder significantly improves the In-File completion baseline by over 10% in all settings and consistently outperforms the vanilla retrieval-augmented code completion approach. Furthermore, we validate the effectiveness of RepoCoder through comprehensive analysis, providing valuable insights for future research. Our source code and benchmark are publicly available: https://github.com/microsoft/CodeT/tree/main/RepoCoder

## 1 Introduction

In real-world software production, it is crucial for developers to be aware of other files within the repository during programming. This challenge gives rise to the task of *repository-level code completion*, where automated tools are expected to utilize the broader context of a repository rather than relying solely on in-file information to complete unfinished code. Code files within a repository often exhibit interrelated dependencies, including shared utilities, configurations, and cross-API invocations resulting from modularization (Tu et al.,

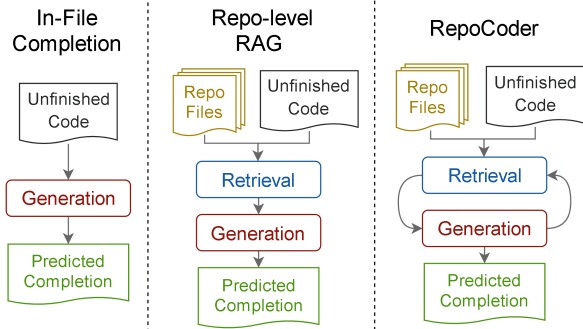

Figure 1: Illustration of the In-File code completion method, the repository-level Retrieval-Augmented Generation (RAG) method, and the iterative retrieval-generation RepoCoder method.

2014). Additionally, each repository typically follows customized naming conventions and coding styles (Zou et al., 2019), which contribute to enhanced readability and maintainability. However, developing effective repository-level code completion tools remains an open problem. Although approaches relying on static code analysis and heuristic rules (Raychev et al., 2014; Svyatkovskiy et al., 2019, 2021) can reliably parse specific repository context, they have limitations in the completion scenario, limiting capability for varying-length completions anywhere in a file. Meanwhile, studies (Hellendoorn and Devanbu, 2017; Svyatkovskiy et al., 2020; Ding et al., 2022) tuning language models on labeled data excel in their respective evaluation scenarios but face challenges generalizing to unseen repositories without retraining.

In this paper, we propose an approach to leverage off-the-shelf retrievers in order to locate valuable information within a repository and enhance the context for language models. We introduce a novel framework called RepoCoder that aims to improve code retrieval and completion performance. As depicted in Figure 1, we enhance the conventional In-File code completion method by incorporating the Retrieval-Augmented Generation

(RAG) technique, which allows us to search for relevant code snippets from the repository to assist in generating the code completion. Additionally, we introduce RepoCoder, which employs an iterative pipeline that utilizes the generated code completion to enhance the retrieval process, thus bridging the gap between the retrieval context and the intended completion target. Figure 2 provides an example that illustrates the rationale behind our design. We demonstrate that relying solely on the unfinished code is insufficient to retrieve useful information from the repository. In the example, the model improvises a statement calling the COLMAP API in the first iteration. The predicted parameters are reasonable yet incorrect. This is because the incomplete code preceding the code completion does not serve as an adequate retrieval query for the intended completion target. However, by performing a subsequent retrieval from the repository using the model's generated completion, we can successfully retrieve the target API signature and complete the code effectively.

Furthermore, we introduce the RepoEval benchmark designed for evaluating the repository-level code completion task, which is constructed using the latest high-quality repositories sourced from GitHub[1]. By introducing RepoEval, we address the lack of established benchmarks in the repository-level scenario. Notably, RepoEval is the first benchmark that encompasses three levels of code completion granularity: line, API invocation, and function body. We also leverage unit tests present in the repository to enhance the accuracy of evaluation, which overcomes the limitations of similarity-based metrics. To rigorously validate the effectiveness of RepoCoder, we conduct extensive experiments using different language models of varying sizes, including GPT-3.5-Turbo[2] and CODE-GEN (Nijkamp et al., 2022). Experimental results demonstrate that RepoCoder achieves significant improvements over In-File completion performance, surpassing the baseline by over 10% across different experimental settings. Moreover, our iterative framework consistently enhances the performance of vanilla retrieval-augmented generation. We also provide a comprehensive analysis of the effectiveness and limitations of RepoCoder, offering insights for future research. Our contributions can be summarized as follows:

[1] https://github.com
[2] https://platform.openai.com/docs/models/gpt-3-5

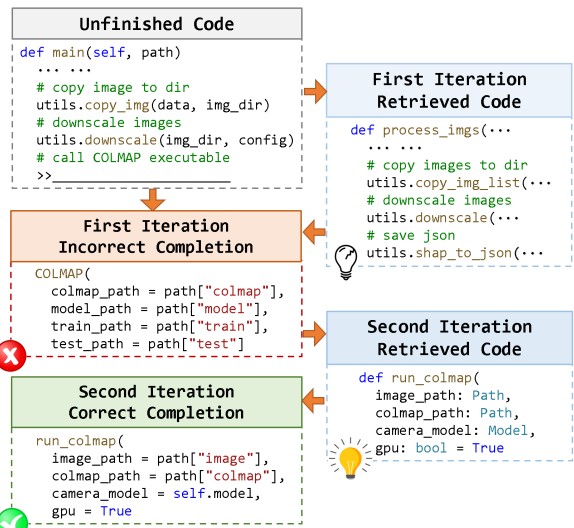

Figure 2: A motivating example showcasing the utilization of model predictions to enhance the performance of code retrieval.

- We propose RepoCoder, a novel iterative retrieval-generation framework for the repository-level code completion task.

- We introduce the RepoEval benchmark, enabling the evaluation of repository-level code completion with varying levels of granularity and improved evaluation accuracy through the utilization of unit tests.

- Through rigorous experimentation, we demonstrate that RepoCoder significantly outperforms the In-File code completion paradigm and enhances the performance of vanilla retrieval-augmented generation.

## 2 Methodology

### 2.1 Overall Framework

The task of code completion using a language model $\mathcal{M}$ can be generally described as $\hat{Y} = \mathcal{M}(X)$, where $\hat{Y}$ represents the predicted tokens and $X$ corresponds to the in-file unfinished code. By introducing an additional code retrieval model $\mathcal{R}$, we can transform the code completion pipeline into a Retrieval-Augmented Generation (RAG) approach. Initially, we establish a retrieval database by partitioning the code files from the repository into a collection of code snippets $C_{repo} = \{c_1, c_2, \cdots\}$. Subsequently, we utilize the retrieval model $\mathcal{R}$ to extract the most relevant code snippets from $C_{repo}$ by employing the unfinished code $X$

as the retrieval query. This process yields a set of retrieved code snippets $C_{ret} = \mathcal{R}(C_{repo}, X)$. Following this, we leverage the language model $\mathcal{M}$ to perform code completion, resulting in the prediction $\hat{Y} = \mathcal{M}(C_{ret}, X)$. Consequently, we are able to incorporate the contextual information from the repository level during the code completion task.

However, using the unfinished code $X$ as the sole retrieval query introduces a gap between the retrieval context and the intended completion target, as exemplified in Figure 2. To address this limitation, we propose RepoCoder, an iterative retrieval-generation pipeline designed to further enhance the performance of the vanilla RAG method. Specifically, for the $i$-th retrieval-generation ($i > 1$) iteration, RepoCoder utilizes the previous model prediction $\hat{Y}^{i-1}$ to construct a new query for the retrieval process. This leads to the generation of another set of relevant code snippets $C_{ret}^i = \mathcal{R}(C_{repo}, X, \hat{Y}^{i-1})$. Subsequently, a new prompt is constructed using $C_{ret}^i$, resulting in the generation of a new prediction $\hat{Y}^i = \mathcal{M}(C_{ret}^i, X)$. The newly generated code completion can serve as either the output of RepoCoder or be utilized for the subsequent retrieval-generation iteration.

Importantly, it is worth noting that the parameters of $\mathcal{M}$ and $\mathcal{R}$ remain unchanged throughout the entire process. Moreover, there is no requirement for static code analysis tools or heuristic rules to construct the retrieval database. In the following subsections, we provide a detailed explanation of the code retrieval process (Section 2.2) and the code generation process (Section 2.3).

## 2.2 Code Retrieval

The retriever utilized within the RepoCoder framework can be any model capable of searching for relevant documents given a specific query. To construct the retrieval database, a *sliding window* approach is employed. The sliding window traverses the files in the repository and extracts contiguous lines of code that fit within the window size, denoted as $S_w$. The sliding window moves a fixed number of lines at each iteration, which is referred to as the sliding size, denoted as $S_s$.

During the initial retrieval process, when no model prediction is available, the query is formulated using the last $S_w$ lines of the unfinished code $X$. Consequently, the most similar code snippets are retrieved using the retrieval model, resulting in $C_{ret}^1 = \mathcal{R}(C_{repo}, X)$. However, a gap exists

```
# Below are some referential code fragments       Retrieved
from other files:                                    Code
# --------------------------------------------
# the below code fragment can be found in:
# tests/test_pipelines_common.py
# --------------------------------------------
# @unittest.skipIf(torch_device != "cuda")
# def test_to_device(self):
#     components = self.get_dummy_components()
#     pipe = self.pipeline_class(**components)
#     pipe.progress_bar(disable=None)
#     pipe.to("cpu")
# --------------------------------------------
"""Based on above, complete the following code:"""
```
```
@unittest.skipIf(torch_device != "cuda")          Unfinished
def test_float16_inference(self):                    Code
    components = self.get_dummy_components()

    pipe = self.pipeline_class(**components)        Model
    pipe.to(torch_device)                         Prediction
```

Figure 3: A visual example demonstrating the format of the RepoCoder prompt, which combines the retrieved code snippets from the repository with the unfinished code present in the target file.

between the retrieval context, based on $X$, and the intended completion target, which is to continue writing $X$. A possible solution is to adjust $C_{ret}^1$ by shifting each code snippet down by a few lines to include the subsequent code. Although this shifting approach has shown effectiveness in previous work (Lu et al., 2022), indiscriminately shifting all retrieved code snippets without considering their content may not always be appropriate.

To address this issue, RepoCoder augments the retrieval query during the $i$-th iteration ($i > 1$) with the previously generated code $\hat{Y}^{i-1}$. Despite the lack of customized information for new repositories, pre-trained code language models have demonstrated impressive general-domain understanding and generation capabilities. The generated code $\hat{Y}^{i-1}$ can provide valuable supplementary information for the retrieval process, even though its correctness may not be guaranteed. Therefore, for the $i$-th iteration of retrieval ($i > 1$), the query is constructed by concatenating the last $(S_w - S_s)$ lines of $X$ with the first $S_s$ lines of $\hat{Y}^{i-1}$. This approach yields the grounded retrieval results $C_{ret}^i = \mathcal{R}(C_{repo}, X, \hat{Y}^{i-1})$.

## 2.3 Code Generation

The generator employed within the RepoCoder framework can be any pre-trained language model capable of predicting subsequent tokens given a specific prompt. As mentioned earlier, it is crucial to incorporate both the context from the repository $C_{repo}$ and the context within the target file for effective code completion. This enables the model to

| ID | Name | License | Created | F. | N. |
|----|------|---------|---------|----|----|
| \multicolumn{6}{c}{Function Body Completion Dataset} |||||| |
| 1. | imagen | MIT License | 2022-05-23 | 14 | 67 |
| 2. | tracr | Apache V2.0 | 2022-12-01 | 56 | 146 |
| 3. | lightmmm | Apache V2.0 | 2022-02-10 | 36 | 64 |
| 4. | inspection | Apache V2.0 | 2022-05-05 | 16 | 32 |
| 5. | omnivore | CC BY-NC 4.0 | 2022-01-20 | 66 | 22 |
| 6. | redframes | BSD-2-Clause | 2022-08-21 | 49 | 42 |
| \multicolumn{6}{c}{Line and API Invocation Completion Datasets} |||||| |
| 7. | rl | MIT License | 2022-02-01 | 165 | 400 |
| 8. | ACE | Apache V2.0 | 2022-11-23 | 425 | 400 |
| 9. | vizier | Apache V2.0 | 2022-02-16 | 188 | 400 |
| 10. | fortuna | Apache V2.0 | 2022-11-17 | 168 | 400 |
| 11. | evaluate | Apache V2.0 | 2022-03-30 | 180 | 400 |
| 12. | diffusers | Apache V2.0 | 2022-05-30 | 305 | 400 |
| 13. | nerfstudio | Apache V2.0 | 2022-05-31 | 157 | 400 |
| 14. | FedScope | Apache V2.0 | 2022-03-24 | 443 | 400 |

Table 1: The repositories utilized in our RepoEval benchmark, presenting statistics obtained from the Github API as of January 2023. *ID* denotes the repository IDs. *F.* indicates the total number of Python source files. *N.* represents the number of extracted test samples from each repository. For brevity, additional repository information can be found in Appendix A.

leverage grounding information and enhances its generalization ability to unseen repositories.

In the RepoCoder framework, we retrieve the most relevant code examples, denoted as $C_{ret}$, from the repository and concatenate them with the unfinished code $X$. To ensure readability and comprehension, we create a prompt template that seamlessly integrates $X$ and $C_{ret}$, as illustrated in Figure 3. The retrieved code snippets are arranged in ascending order based on their similarity scores to the query. Each code snippet is accompanied by its original file path, and the maximum number of code snippets included in the prompt, denoted as $K$, depends on the available prompt length. Ultimately, the prompt contains as much relevant information as possible to facilitate code completion.

## 3 Benchmark Construction

To facilitate the evaluation of code completion tools in the repository-level scenario, we propose a novel RepoEval benchmark. This benchmark is carefully constructed using the latest high-quality repositories sourced from GitHub and encompasses three levels of code completion granularity: line, API invocation, and function body. To assess the correctness of completed functions, we utilize unit tests present in the repository instead of relying solely on similarity-based metrics. Each sample in the RepoEval benchmark is annotated with the corre-

sponding source repository, file path, line numbers, and ground truth completion. For analysis and unit test execution, complete copies of the repositories are archived as of January 2023.

To construct RepoEval, we first meticulously curate a collection of Python repositories from GitHub that satisfy the following criteria: open-source license, created after January 1, 2022[3], non-fork original repositories, over 100 stars, over $80\%$ of files written in Python, and explicit unit tests. Furthermore, to mitigate potential biases, we employ a random selection process for the repositories and create three distinct datasets for line completion, API invocation completion, and function body completion. Additional details regarding the selected repositories can be found in Table 1.

**Line completion:** In adherence to the conventions of code completion benchmarks (Lu et al., 2021, 2022), we implement the line completion scenario. First, according to the above-mentioned criteria, we select 8 repositories that vary in size and cover different domains. Then we randomly select 200 lines to complete from each repository, ensuring the lines are non-repetitive, not code comments, and each line contains at least 5 tokens. Eventually, a total of 1600 test samples are generated for the line completion dataset.

**API Invocation Completion:** We also choose to test the API completion scenario, especially in-repository defined APIs. It is a harder problem than the completion of built-in or third-party APIs due to the lack of customized training data (Hellendoorn et al., 2019). We utilize the same group of repositories in the line dataset and parse the target repositories to locate invocations of in-repository APIs. From these candidates, we then randomly select 200 non-repetitive API invocations from each repository, resulting in a total of 1600 test samples for the API invocation completion dataset.

**Function Body Completion:** Alongside the line and API completion evaluations, we also assess the ability to complete function bodies, which requires executing unit tests present in the repository. However, running tests can be time-consuming and computationally expensive. To address this, we randomly select a separate set of smaller-scale repositories that are easy to deploy. Within these repositories, we locate functions covered by unit tests

---

[3]The training data of GPT-3.5-Turbo and CODEGEN is up to 2021. We use data from 2022 to prevent data leakage.

and select function bodies containing 3 to 30 lines of code to complete. This yields a total of 373 test samples for the function body completion dataset.

# 4 Experimental Setup

## 4.1 Methods for Comparison

**In-File Completion:** Previous studies (Chen et al., 2021; Nijkamp et al., 2022; Chen et al., 2022) have demonstrated the effectiveness of utilizing large pre-trained language models for code generation in a zero-shot completion manner, conditioned on the provided context. Furthermore, it has been established that incorporating in-file context is beneficial for code completion scenarios (Clement et al., 2021). Hence, as a baseline, we implement an In-File completion method by populating the prompt with the unfinished code and directly utilizing the pre-trained code generation model to predict the code completion.

**Oracle Method:** A key contribution of RepoCode is the integration of model predictions for retrieval, bridging the gap between retrieval and the intended completion target. To showcase the effectiveness of this approach, we devise an oracle retrieval-augmented generation method for comparison purposes. This method performs a single retrieval process to obtain relevant code snippets, denoted as $C_{ret}^{gt}$, by utilizing the last $S_w - S_s$ lines of $X$ and the first $S_s$ lines of the ground truth code, $Y$. Subsequently, the completion code, denoted as $\hat{Y}$, is generated through $\mathcal{M}(C_{ret}^{gt}, X)$. This allows us to achieve the upper bound of performance for RepoCoder, conditioned on the retrieval model $\mathcal{R}$ and the generation model $\mathcal{M}$.

## 4.2 Implementation Details

**Retrieval Model:** For our main experiments, we employ a sparse bag-of-words model as the retrieval model, which has demonstrated effectiveness in retrieving similar code snippets (Lu et al., 2022). This model transforms the query and candidate code snippets into sets of tokens and calculates their similarity using the Jaccard index (Jaccard, 1912), computed as $Jaccard(S_q, S_c) = \frac{|S_q \cap S_c|}{|S_q \cup S_c|}$, where $S_q$ and $S_c$ represent the tokens of the query and candidate code snippets, respectively. We also experiment with a dense retriever based on UniXcoder (Guo et al., 2022), detailed in Appendix B.

**Generation Model:** We evaluate RepoCoder using four pre-trained language models with varying code generation capabilities. The first model, GPT-3.5-Turbo, is a state-of-the-art commercial code generation model with billions of trainable parameters and has been pre-trained on an extensive code corpus. Access to GPT-3.5-Turbo is obtained through the API provided by OpenAI. The second model, CODEGEN, is an open-source code generation model that has multiple published versions with varying model sizes and training data. In our experiments, we utilize three versions of CODEGEN model with 6B, 2B, and 350M parameters.

**Hyper-parameters:** We found that RepoCoder's performance was not highly sensitive to changes in hyper-parameters. Therefore, for our experiments on RepoEval, we assign hyper-parameters based on our experience. Specifically, the maximum number of tokens for the combined input prompt and output prediction is set to $4,096$ for GPT-3.5-Turbo and $2,048$ for CODEGEN. The length of retrieved code snippets is set to half the prompt length. For line and API completion, the maximum number of tokens in the generated completion ($\hat{Y}$), the line length of the sliding window ($S_w$), and the sliding size ($S_s$) are set to 100, 20, and 10 respectively. For function body completion, these values are adjusted to 500, 50, and 10. The maximum number of retrieved snippets ($K$) is set to 10. The same hyper-parameters were used for the single-iteration RAG, iterative RepoCoder, and Oracle baselines, ensuring a fair comparison between methods. Notably, given that these parameters are intricately linked to the programming language and contextual scenarios, practitioners should make adjustments to ensure optimal real-world performance.

## 4.3 Evaluation Metrics

**Similarity-based Evaluation:** Following established practices in code completion research (Lu et al., 2021, 2022), we evaluate our line and API completion datasets using two metrics: Exact Match (EM) and Edit Similarity (ES). The EM score is a binary metric that takes the value of 1 if the predicted code exactly matches the ground truth code, and 0 otherwise. The ES metric provides a more fine-grained evaluation and is calculated as $ES = 1 - \frac{\text{Lev}(\hat{Y}, Y)}{\max(|\hat{Y}|, |Y|)}$, where Lev represents the Levenshtein distance (Levenshtein et al., 1966).

**Execution-based Evaluation:** For the function body completion dataset, we utilize unit tests present in the repository to evaluate functional

**Table 2a: Line Completion**

| Metric | Oracle | In-File | RepoCoder Iterations | | | |
|---|---|---|---|---|---|---|
| | | | 1 | 2 | 3 | 4 |
| GPT-3.5-Turbo | | | | | | |
| EM | 57.75 | 40.56 | 55.31 | 56.81 | **57.00** | 56.63 |
| ES | 75.43 | 65.06 | 74.38 | 75.11 | **75.30** | 75.10 |
| CODEGEN-MONO-6B | | | | | | |
| EM | 48.81 | 34.56 | 45.81 | 47.06 | **47.75** | 47.44 |
| ES | 71.02 | 60.67 | 69.21 | 70.10 | **70.73** | 70.19 |
| CODEGEN-MONO-2B | | | | | | |
| EM | 47.31 | 33.63 | 44.56 | 46.94 | 46.69 | **47.13** |
| ES | 69.80 | 58.99 | 67.68 | 68.82 | 68.62 | **68.92** |
| CODEGEN-MONO-350M | | | | | | |
| EM | 45.19 | 29.56 | 41.88 | 43.06 | **43.94** | 43.06 |
| ES | 67.20 | 55.39 | 65.05 | 65.66 | **65.97** | 65.62 |

(a) Line Completion.

**Table 2b: API Invocation Completion**

| Metric | Oracle | In-File | RepoCoder Iterations | | | |
|---|---|---|---|---|---|---|
| | | | 1 | 2 | 3 | 4 |
| GPT-3.5-Turbo | | | | | | |
| EM | 50.13 | 34.06 | 47.69 | 49.19 | 49.44 | **49.56** |
| ES | 74.50 | 63.22 | 73.63 | 74.43 | **74.59** | 74.48 |
| CODEGEN-MONO-6B | | | | | | |
| EM | 40.25 | 26.19 | 36.69 | 38.88 | 39.13 | **39.31** |
| ES | 67.94 | 56.45 | 64.20 | 65.52 | 65.53 | **65.90** |
| CODEGEN-MONO-2B | | | | | | |
| EM | 39.44 | 25.44 | 35.44 | 37.56 | **38.44** | 38.25 |
| ES | 66.78 | 56.88 | 63.47 | 64.15 | 64.53 | **64.60** |
| CODEGEN-MONO-350M | | | | | | |
| EM | 34.88 | 22.19 | 31.75 | **33.88** | 33.75 | 33.81 |
| ES | 63.06 | 52.24 | 59.82 | 61.03 | 60.96 | **61.06** |

(b) API Invocation Completion.

Table 2: Performance comparison on the line and API invocation completion datasets. Results present the average performance of each method evaluated using Exact Match (EM) and Edit Similarity (ES) scores. Numbers are shown in percentage (%), with the best performance highlighted in bold.

correctness. This approach is more reliable than similarity-based metrics in assessing the behavior of the completed functions. While collecting unit tests can be time-consuming, we focus on a realistic scenario and utilize the unit tests available in GitHub repositories to validate the generated code. We execute the completed code and report the Pass Rate (PR), where PR is 1 if the code passes all the corresponding test cases, and 0 otherwise.

## 5 Experimental Results

### 5.1 Line and API Completion Datasets

We compare the performance of RepoCoder with the In-File completion method and the Oracle method on the line and API invocation completion datasets using four pre-trained language models and different retrieval-generation iterations. From the results listed in Table 2a and 2b, we find that RepoCoder consistently improves the In-File completion performance on both datasets across all model sizes. The absolute improvements in the Exact Match (EM) and Edit Similarity (ES) scores exceed 10% and 8%, respectively. RepoCoder also shows competitive results compared to the Oracle method. With two or more iterations, RepoCoder consistently outperforms the vanilla Retrieval-Augmented Generation (RAG) approach for all language models. Additionally, the CODE-GEN model with 350M parameters is comparable to the GPT-3.5-Turbo model with In-File completion when integrated with RepoCoder. We also test RepoCoder using a dense retriever powered

| ID | N. | Oracle | In-File | RepoCoder Iterations | | | |
|---|---|---|---|---|---|---|---|
| | | | | 1 | 2 | 3 | 4 |
| 1. | 67 | 56.72 | 29.85 | 53.73 | **55.22** | 55.22 | 55.22 |
| 2. | 146 | 43.84 | 27.40 | 41.78 | 43.84 | **44.52** | 44.52 |
| 3. | 64 | 32.81 | 10.94 | 25.00 | **34.38** | 31.25 | 32.81 |
| 4. | 32 | 34.38 | 28.13 | 34.38 | **37.50** | 34.38 | 34.38 |
| 5. | 22 | 40.91 | 31.82 | 31.82 | **36.36** | 31.82 | 36.36 |
| 6. | 42 | 38.10 | 9.52 | 28.57 | **38.10** | 38.10 | 38.10 |
| All | 373 | 42.63 | 23.32 | 38.34 | **42.63** | 41.82 | 42.36 |

Table 3: Performance comparison on the function body completion dataset using GPT-3.5-Turbo. Results display the Pass Rate (PR) of each method as evaluated using test cases. Numbers are presented in percentage (%), with the best performance highlighted in bold. *ID* represents the repository IDs, and *N.* indicates the number of test samples in each repository.

by UniXcoder (Guo et al., 2022) (detailed in Appendix B) and find that the simple sparse retriever achieves equivalent performance, highlighting the robustness of RepoCoder across different code retrieval and generation models.

### 5.2 Function Completion Dataset

We proceed to assess the performance of RepoCoder on the function body completion dataset. To tackle the greater difficulty of function body completion, we employ the GPT-3.5-Turbo model due to its superior code understanding and generation capabilities, as well as its larger prompt length suitable for longer function code snippets. The evaluation results, presented in Table 3, showcase similar trends to our findings on the line and API invo-

| Metric | GT-Code | In-File | RepoCoder Iter- | |
| --- | --- | --- | --- | --- |
| | | | 1 | 2 |
| GPT-3.5-Turbo | | | | |
| EM | 55.54 | 34.42 | 53.63 | 55.07 |
| ES | 77.67 | 62.75 | 77.68 | 78.40 |
| Recall | 100.0 | - | 86.04 | 90.34 |
| CODEGEN-MONO-6B | | | | |
| EM | 44.78 | 26.87 | 41.09 | 44.04 |
| ES | 71.47 | 56.42 | 67.10 | 68.55 |
| Recall | 100.0 | - | 76.27 | 82.92 |
| CODEGEN-MONO-350M | | | | |
| EM | 37.86 | 22.25 | 35.64 | 38.13 |
| ES | 66.20 | 52.40 | 62.82 | 64.26 |
| Recall | 100.0 | - | 76.27 | 80.89 |

Table 4: Performance comparison on the test samples extracted from the API completion dataset using GT-Code, In-File, and RepoCoder methods. Results present the averaged performance of each method as evaluated by Exact Match (EM), Edit Similarity (ES), and Recall. Numbers are shown in percentage (%).

| Method | Oracle | | RepoCoder Iter-2 | |
| --- | --- | --- | --- | --- |
| Dataset | Line | API | Line | API |
| Location Statistics | | | | |
| Imported | 4.22% | 8.16% | 3.22% | 9.06% |
| Current File | 3.86% | 4.05% | 3.32% | 4.10% |
| Current Directory | 46.41% | 58.40% | 45.68% | 59.58% |
| Similar Import | 82.15% | 86.71% | 82.80% | 87.70% |
| Similar Name | 52.23% | 65.10% | 53.40% | 64.20% |
| Others | 7.27% | 4.43% | 7.77% | 3.35% |
| Eligible Samples | | | | |
| Test Samples | 333 | 294 | 312 | 276 |
| Code Snippets | 2202 | 1851 | 2047 | 1732 |

Table 5: Locations of retrieved code snippets when the Oracle/RepoCoder method outperforms the In-File completion method using GPT-3.5-Turbo on the line and API completion datasets.

cation completion datasets. Across most repositories, RepoCoder exhibits significant improvement over the In-File completion method and competitive performance compared to the Oracle method. Moreover, with additional retrieval-generation iterations, RepoCoder consistently outperforms the vanilla Retrieval-Augmented Generation (RAG) approach. These results reaffirm the effectiveness of our approach.

# 6 Analysis

In this section, we conduct further analyses on the retrieved code snippets to gain a deeper understanding of RepoCoder and provide valuable insights for future research.

## 6.1 Quality of Retrieved Code

We observe a significant impact of the retrieved code's quality on code completion performance. And the most helpful code snippets typically contain code statements similar to the target completion or demonstrate example usages of the target API invocation. Then, to validate the correlation between retrieval quality and completion performance, we design an analysis experiment using the API invocation completion dataset. In this experiment, we leverage a static code analysis tool to locate code snippets in other files that include invocations of the ground truth API. Subsequently, we rank these code snippets based on their similarity to the unfinished code and select the most similar

ones to include in the completion prompt. We refer to this method as GT-Code and compare its performance against the In-File and RepoCoder methods. Additionally, we show the recall performance of RepoCoder by counting the number of retrieved code snippets containing invocation examples of the ground truth API.

Since not every API in the test dataset has invocation examples in other files, and we also exclude the invocation examples existing in the input prompt for the model, we finally extract from the API invocation dataset 1046 and 1083 eligible test samples respectively for the GPT-3.5-Turbo and CODE-GEN models to conduct the experiment. From the obtained results in Table 4, we observe that the GT-Code method, which utilizes ground truth API invocation examples, generally achieves the best performance among all methods. Furthermore, RepoCoder with two iterations exhibits higher recall for ground truth API invocations compared to a single-iteration, which likely contributes to its superior code completion performance. Notably, as the language model grows more powerful, the recall value using RepoCoder Iter-2 also increases, indicating the model predictions indeed assist the retrieval process and emphasizing the effectiveness of RepoCoder.

## 6.2 Locations of Retrieved Code

The retrieval of code snippets provides valuable contextual information from other files to enhance the context for language models. We conduct a separate experiment to study the various locations from which effective retrieval occurs. Specifically, we select test samples that are successfully pre-

dicted by the Oracle/RepoCoder method but not by In-File completion using GPT-3.5-Turbo. This yields a number of eligible test samples and retrieved code snippets for the line and API invocation completion datasets. To determine the original source of these code snippets, we adopt a classification scheme inspired by Shrivastava et al. (2022), consisting of five distinct file locations: 1. Imported: code from a file imported by the target file. 2. Current File: code from the excluded content of the target file. 3. Current Directory: code from a file in the same directory as the target file. 4. Similar Import: code from a file sharing at least one same API import with the target file. 5. Similar Name: code from a file with a file name sharing at least one token with the target file (assuming snake-case style file names).

The results are as outlined in Table 5. Our findings indicate a similar distribution of retrieved code snippets between the Oracle method and RepoCoder. The majority of code snippets fall within our defined categories, and a significant portion of code snippets originates from files with "Similar Import", "Similar Name", or "Current Directory" locations, underscoring the importance of contextual information in code completion tasks. Furthermore, we conduct an ablation study, wherein we restrict the retrieval process to only the aforementioned file locations. The results reveal a degradation in performance, highlighting the efficacy and simplicity of RepoCoder in the retrieval process.

## 7 Related Work

**Repository Context in Code Completion:** Incorporating repository-level context into code completion tools has been a long-standing challenge. Traditional code completion techniques typically involve analyzing code to identify potential suggestions, followed by re-ranking them (Raychev et al., 2014; Svyatkovskiy et al., 2019, 2021). While this approach offers efficient performance, it lacks the flexibility to generate code at arbitrary granularity. Another line of research treats code completion as a language modeling task, where the next tokens are generated based on the given context. Several methods have been proposed to incorporate repository context into the training of language models, including n-grams (Tu et al., 2014), LSTMs (Hellendoorn and Devanbu, 2017), and Transformers (Svyatkovskiy et al., 2020; Liu et al., 2022; Ding et al., 2022). However, the process of collecting

labeled data and fine-tuning models for different applications remains resource-intensive. In recent years, there has been significant attention on Large Language Models (LLMs) for code completion. A study by Shrivastava et al. (2022) also explores the scenario of repository-level code completion. Despite its innovative approach, the study relies on inflexible heuristics and classifier training for prompt construction. This highlights the ongoing challenges in effectively leveraging LLMs for code completion and the need for further research.

**Joint Modeling Retrieval and Generation:** Despite the impressive capabilities of LLMs (Brown et al., 2020; Thoppilan et al., 2022; Chowdhery et al., 2022), their offline training paradigm often limits access to customized and up-to-date information. Recent studies have started exploring the joint modeling of retrieval and generation in knowledge-intensive tasks, such as question answering (Guu et al., 2020; Lewis et al., 2020; Izacard et al., 2022) and dialogue generation (Zhang et al., 2022). This approach has also been extended to code generation by incorporating retrieved documents or code examples into the generation process (Rizwan Parvez et al., 2021; Zhou et al., 2022; Lu et al., 2022; Zan et al., 2022). As language models have become increasingly sophisticated, there is a growing trend towards in-context joint retrieval and generation, treating the LLM as a fixed black box (Levine et al., 2022; Ram et al., 2023; Shi et al., 2023). Moreover, some studies have investigated utilizing the model's predictions as supplementary context to inform the retrieval process (Mao et al., 2020; Li et al., 2022; Wang et al., 2023; Zemlyanskiy et al., 2022). In this work, we demonstrate that adopting an iterative paradigm that combines code retrieval and generation can serve as an effective method for repository-level code completion.

## 8 Conclusion and Future Work

In conclusion, we introduce RepoCoder, a straightforward and effective framework for the repository-level code completion task. By leveraging a retriever and a language model, RepoCoder effectively utilizes repository-level information. Through an iterative process of retrieval and generation, RepoCoder bridges the gap between retrieval context and the target code, resulting in improved code completion performance. Our extensive experiments conducted on the RepoEval benchmark demonstrate that RepoCoder consistently and

significantly enhances In-File completion performance, surpassing the vanilla Retrieval-Augmented Generation (RAG) approach. Furthermore, our analysis provides valuable insights into the rationale and limitations of RepoCoder. With its simplicity, versatility, and effectiveness, RepoCoder has the potential to become an indispensable tool in real-world software development, and we aim to further enhance its usability and robustness.

## Limitations

**Limited Effectiveness for Repositories with Low Code Duplication:** Despite we have demonstrated the effectiveness of RepoCoder through extensive experiments and analysis, RepoCoder may not bring significant performance improvements when a repository has few instances of code duplication. In such scenarios, the code retrieval process struggles to find sufficient relevant information from the repository to facilitate code completion. This issue is further highlighted in the study presented in Appendix C.

**Difficulty in Identifying the Optimal Number of Iterations:** While RepoCoder with two iterations outperforms the RAG method, determining the optimal number of iterations remains a challenge. Subsequent iterations of RepoCoder may exhibit unstable performance compared to previous iterations. Appendix D provides a demonstration of this issue. To mitigate this, we have explored different approaches to automatically terminate the iteration process when necessary. However, finding an optimal stopping criterion without significantly impacting RepoCoder's performance remains an ongoing challenge. Further research is required to develop techniques that can identify the iteration at which RepoCoder achieves the best performance.

**Time Efficiency for Real-Time Deployment:** While RepoCoder demonstrates promising gains in code completion accuracy through iterative retrieval and generation, concerns may arise due to the latency of additional retrieval-generation steps. For real-time deployment scenarios with strict latency requirements, we can further improve RepoCoder through model optimizations such as quantization, distillation, and hardware acceleration to expedite inference. Techniques like caching frequent code snippets and pre-processing repositories can also boost speed. The model iterations can be dynamically adapted based on latency goals and contextual needs to balance accuracy and efficiency. Nevertheless, improving time efficiency is another important topic that is out of the scope of our current paper.

**Limited Exploration of Different Experimental Settings:** First, while we have validated the effectiveness of RepoCoder, we have not yet explored the potential improvements that can be achieved through different prompt templates. We believe that more careful prompt engineering could enhance the performance of our approach even further. Second, our focus in this study has primarily been on exploring similarity-based retrieval models. The reason for this limited scope is rooted in the complexity of code retrieval, which involves numerous intricate details that are not directly relevant to the RepoCoder framework. Considering alternative retrieval models or expanding the exploration to other code retrieval techniques could provide further insights and comparative evaluations. Third, we have observed significant advancements in code generation models, such as GPT-4 (OpenAI, 2023), StarCoder (Li et al., 2023), and WizardCoder (Luo et al., 2023). While our experiments demonstrate the efficacy of RepoCoder across different language models (GPT-3.5-Turbo and CODEGEN), it would be valuable to investigate how our approach performs with these advanced code generation models. Incorporating them into our experimental setup would provide a broader evaluation of RepoCoder across a wider range of language models. Fourth, our experiments primarily use the In-File and Oracle methods as baselines. This decision stems from the fact that repository-level code completion using language models is a relatively new task, lacking well-established and reproducible baselines. To provide further insights, we include comparisons to other commercial code completion products. Nonetheless, it is impractical to systematically benchmark against complex, confidential commercial products. We instead conduct a study in Appendix E showcasing the repository-level completion ability of RepoCoder and another three major commercial products, where we can illustrate their qualitative differences. In summary, future work should aim to explore different prompt designs, consider alternative retrieval or generation models, and incorporate additional baselines.

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

# A  Repository Details

As mentioned in Section 3, we meticulously selected repositories for our RepoEval benchmark based on criteria such as open-source license, creation date, code quantity, and quality. Detailed information about these repositories is provided in Table 7.

# B  Using the Dense Retriever

In our main experiments (as described in Section 4.2), we utilize a sparse retrieval model for RepoCoder due to its acceptable performance and computational efficiency. However, RepoCoder is a versatile framework that can be applied with other code retrieval models as well. To further validate the effectiveness of RepoCoder, we conduct additional experiments using a dense code retriever.

Specifically, we employ UniXcoder (Guo et al., 2022), a state-of-the-art code embedding model, to

| Metric | Oracle | In-File | RepoCoder Iterations | | | |
|---|---|---|---|---|---|---|
| | | | 1 | 2 | 3 | 4 |
| GPT-3.5-Turbo | | | | | | |
| EM | 57.25 | 40.56 | 54.56 | 56.25 | **56.31** | 55.31 |
| ES | 75.80 | 65.06 | 73.96 | **74.70** | 74.31 | 74.04 |
| CODEGEN-MONO-6B | | | | | | |
| EM | 47.25 | 34.56 | 45.56 | **46.75** | 46.56 | 46.63 |
| ES | 70.03 | 60.67 | 68.89 | **69.51** | 69.33 | 69.41 |
| CODEGEN-MONO-2B | | | | | | |
| EM | 45.75 | 33.63 | 44.94 | 45.13 | **45.81** | 45.50 |
| ES | 68.35 | 58.99 | 67.43 | 68.01 | **68.35** | 68.20 |
| CODEGEN-MONO-350M | | | | | | |
| EM | 42.88 | 29.56 | 41.50 | **42.69** | 42.13 | 42.31 |
| ES | 65.97 | 55.39 | 64.71 | **65.64** | 65.13 | 65.53 |

(a) Line Completion.

| Metric | Oracle | In-File | RepoCoder Iterations | | | |
|---|---|---|---|---|---|---|
| | | | 1 | 2 | 3 | 4 |
| GPT-3.5-Turbo | | | | | | |
| EM | 50.06 | 34.06 | 47.56 | 49.13 | 49.19 | **49.63** |
| ES | 74.95 | 63.22 | 72.66 | 74.22 | 72.83 | **74.41** |
| CODEGEN-MONO-6B | | | | | | |
| EM | 39.56 | 26.19 | 37.00 | 38.44 | **39.13** | 38.44 |
| ES | 66.82 | 56.45 | 64.21 | 65.33 | 65.26 | **65.53** |
| CODEGEN-MONO-2B | | | | | | |
| EM | 37.06 | 25.44 | 35.00 | **37.19** | 36.13 | 36.63 |
| ES | 64.69 | 56.88 | 62.22 | 63.39 | 63.17 | **63.65** |
| CODEGEN-MONO-350M | | | | | | |
| EM | 33.13 | 22.19 | 31.50 | 32.69 | **32.75** | 31.94 |
| ES | 61.37 | 52.24 | 59.35 | **60.69** | 60.49 | 60.34 |

(b) API Invocation Completion.

Table 6: Performance comparison on the line and API invocation completion datasets using the dense retriever. Results display the average performance of each method evaluated using Exact Match (EM) and Edit Similarity (ES) scores. Numbers are presented in percentage (%), with the best performance highlighted in bold.

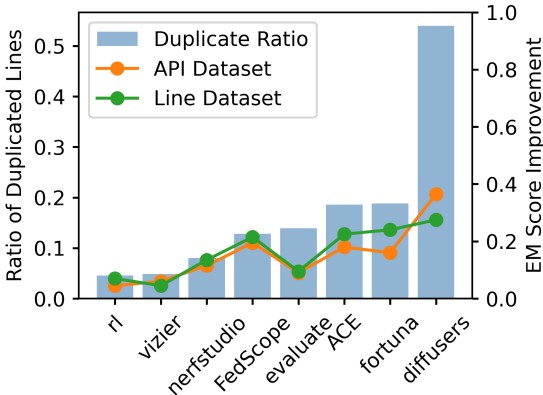

Figure 4: Correlation between the absolute performance improvements achieved by RepoCoder Iter-2 over the In-File method and the repository duplication ratios.

transform code snippets into hidden vectors. We then calculate the similarity between code snippets using cosine similarity. The experimental results on the line and API invocation completion datasets using the dense retriever are presented in Table 6a and Table 6b. Notably, the performance of RepoCoder using the dense retriever is comparable to that using the sparse retriever. Furthermore, the findings remain consistent across both retrievers, highlighting the robustness and generalizability of RepoCoder.

## C Code Duplication in Repositories

We explore the relationship between the performance of RepoCoder and the code duplication ratio of the repositories. Intuitively, since RepoCoder utilizes similarity-based retrieval to find code exemplars, one might expect a positive correlation between its performance and the code duplication ratio. To assess this relationship, we calculate the code duplication ratio of the repositories by determining the ratio of duplicated code lines to the total code lines.

Figure 4 presents the results, demonstrating the correlation between RepoCoder's performance, as measured by the Exact Match (EM) metric, and the code duplication ratio on the line and API completion datasets using GPT-3.5-Turbo. Notably, the repository "diffusers" exhibits the highest duplication ratio, which corresponds to a significant performance improvement for RepoCoder on both datasets. Conversely, "rl" and "vizier" have low duplication ratios, resulting in comparatively lower performance for RepoCoder. However, the correlation between RepoCoder's performance and the code duplication ratio is not absolute. For example, "FedScope" and "evaluate" have similar duplication ratios but show different performance gains for RepoCoder.

## D Failed Cases between Iterations

In Section 5, the evaluation results demonstrate that increasing the number of RepoCoder iterations does not necessarily guarantee performance improvements. To further investigate this issue, we analyze the changes in the number of correct code completions achieved by different methods on the API invocation completion dataset. The prediction is considered correct when the EM score is 1. Table 8 presents the results, showing the counts

| ID | Name | Github Link | License | Created | Stars | F. | L. |
|----|------|-------------|---------|---------|-------|-----|-----|
| | | | Function Body Completion Dataset | | | | |
| 1. | imagen | lucidrains/imagen-pytorch | MIT License | 2022-05-23 | 6, 160 | 14 | 7, 324 |
| 2. | tracr | deepmind/tracr | Apache V2.0 | 2022-12-01 | 284 | 56 | 9, 110 |
| 3. | lightmmm | google/lightweight_mmm | Apache V2.0 | 2022-02-10 | 353 | 36 | 9, 676 |
| 4. | inspection | amazon-science/patchcore-inspection | Apache V2.0 | 2022-05-05 | 304 | 16 | 2, 532 |
| 5. | omnivore | facebookresearch/omnivore | CC BY-NC 4.0 | 2022-01-20 | 459 | 66 | 11, 797 |
| 6. | redframes | maxhumber/redframes | BSD-2-Clause | 2022-08-21 | 283 | 49 | 3, 881 |
| | | | Line and API Invocation Completion Datasets | | | | |
| 7. | rl | pytorch/rl | MIT License | 2022-02-01 | 873 | 165 | 59, 522 |
| 8. | ACE | opendilab/ACE | Apache V2.0 | 2022-11-23 | 299 | 425 | 66, 497 |
| 9. | vizier | google/vizier | Apache V2.0 | 2022-02-16 | 688 | 188 | 43, 393 |
| 10. | fortuna | awslabs/fortuna | Apache V2.0 | 2022-11-17 | 373 | 168 | 18, 738 |
| 11. | evaluate | huggingface/evaluate | Apache V2.0 | 2022-03-30 | 1, 082 | 180 | 21, 418 |
| 12. | diffusers | huggingface/diffusers | Apache V2.0 | 2022-05-30 | 9, 521 | 305 | 98, 181 |
| 13. | nerfstudio | nerfstudio-project/nerfstudio | Apache V2.0 | 2022-05-31 | 3, 151 | 157 | 27, 289 |
| 14. | FedScope | alibaba/FederatedScope | Apache V2.0 | 2022-03-24 | 811 | 443 | 48, 545 |

Table 7: Detailed information of the Github repositories used for RepoEval. *ID* represents the repository IDs. *F.* denotes the total number of Python source files, while *L.* indicates the total number of non-empty Python code lines. Statistics are accurate as of January 2023.

| | | | RepoCoder Iterations | | | | | | |
|--------|-----|-----|-----|-----|-----|-----|-----|-----|-----|
| In-File | $\rightarrow$ | 1 | $\rightarrow$ | 2 | $\rightarrow$ | 3 | $\rightarrow$ | 4 | |
| | | | GPT-3.5-Turbo | | | | | | |
| +545 | $\begin{smallmatrix}-40\\+258\end{smallmatrix}$ | 763 | $\begin{smallmatrix}-46\\+70\end{smallmatrix}$ | 787 | $\begin{smallmatrix}-83\\+90\end{smallmatrix}$ | 794 | $\begin{smallmatrix}-14\\+10\end{smallmatrix}$ | 790 | |
| | | | CODEGEN-MONO-6B | | | | | | |
| +424 | $\begin{smallmatrix}-57\\+220\end{smallmatrix}$ | 587 | $\begin{smallmatrix}-35\\+70\end{smallmatrix}$ | 622 | $\begin{smallmatrix}-12\\+16\end{smallmatrix}$ | 626 | $\begin{smallmatrix}-7\\+10\end{smallmatrix}$ | 629 | |
| | | | CODEGEN-MONO-2B | | | | | | |
| +412 | $\begin{smallmatrix}-64\\+219\end{smallmatrix}$ | 567 | $\begin{smallmatrix}-32\\+66\end{smallmatrix}$ | 601 | $\begin{smallmatrix}-12\\+26\end{smallmatrix}$ | 615 | $\begin{smallmatrix}-16\\+13\end{smallmatrix}$ | 612 | |
| | | | CODEGEN-MONO-350M | | | | | | |
| +352 | $\begin{smallmatrix}-46\\+202\end{smallmatrix}$ | 508 | $\begin{smallmatrix}-25\\+59\end{smallmatrix}$ | 542 | $\begin{smallmatrix}-18\\+16\end{smallmatrix}$ | 540 | $\begin{smallmatrix}-11\\+12\end{smallmatrix}$ | 541 | |

Table 8: The changes in the number of correct code completions achieved using different methods on the API invocation completion dataset.

of correct code completions for each iteration of RepoCoder. We observe that each iteration of RepoCoder both passes cases that the previous iteration failed and fails cases that the previous iteration has passed.

Upon manually examining the failed cases, we have the following observations: Firstly, a majority of failures are caused by misleading retrieved code, which leads to incorrect predictions. For instance, the same API may have different sets of parameters across different files, and the retrieved API usage example can be misleading in such cases. Secondly, the model's predictions are not always suitable for retrieval. This is because the query is constructed using a fixed length of the predicted code, which may include noisy code beyond the initial lines of

helpful code completion. Furthermore, our investigation reveals that many cases in the line and API datasets are actually correct despite being evaluated as incorrect by the EM score. This highlights the importance of considering the actual functionality of the code, rather than solely relying on exact matching, and suggests incorporating unit tests to assess code correctness.

# E  Case Study of Commercial Products

We conduct a study to showcase the repository-level code completion ability of RepoCoder and another three major commercial code completion tools: Github Copilot [4], Tabnine [5], and Amazon CodeWhisperer [6]. The experiment was conducted using the Visual Studio Code IDE, with each product providing completions as a plugin. These products are based on large language models pre-trained on code data and can perform line-level and block-level completion, similar to our study scenario. We selected a simple API invocation example from the RepoEval dataset. The task was to complete the function body for initializing a *StableDiffusionKDiffusionPipeline*, where the prefix in-file context provided little information. As shown in Figure 5, none of the commercial products generated the correct completion. The implementation details of these commercial products are confidential, it is

---

[4] https://github.com/features/copilot
[5] https://www.tabnine.com
[6] https://aws.amazon.com/codewhisperer

```
       diffusers\utils\dummy_torch_and_transformers_and_k_diffusion_objects.py
from ..utils import DummyObject, requires_backends

class StableDiffusionKDiffusionPipeline(metaclass=DummyObject):
    _backends = ["torch", "transformers", "k_diffusion"]

    def __init__(self, *args, **kwargs):
        raise RuntimeError("Not Instantiate StableDiffusionKDiffusionPipeline")
```

(a) Incorrect code completion of Github Copilot.

```
       diffusers\utils\dummy_torch_and_transformers_and_k_diffusion_objects.py
from ..utils import DummyObject, requires_backends

class StableDiffusionKDiffusionPipeline(metaclass=DummyObject):
    _backends = ["torch", "transformers", "k_diffusion"]

    def __init__(self, *args, **kwargs):
        pass

    @requires_backends(_backends)
    def forward(self, *args, **kwargs):
```

(b) Incorrect code completion of Tabnine.

```
       diffusers\utils\dummy_torch_and_transformers_and_k_diffusion_objects.py
from ..utils import DummyObject, requires_backends

class StableDiffusionKDiffusionPipeline(metaclass=DummyObject):
    _backends = ["torch", "transformers", "k_diffusion"]

    def __init__(self, *args, **kwargs):
        requires_backends(self, "k_diffusion")
```

(c) Incorrect code completion of Amazon CodeWhisperer.

```
       diffusers\utils\dummy_torch_and_transformers_and_k_diffusion_objects.py
# the below code fragment can be found in:
# diffusers\utils\dummy_torch_and_transformers_and_k_diffusion_objects.py
# ------------------------------------------------------
# class AltDiffusionImg2ImgPipeline(metaclass=DummyObject):
#     _backends = ["torch", "transformers"]
#     def __init__(self, *args, **kwargs):
#         requires_backends(self, ["torch", "transformers"])
# ------------------------------------------------------
from ..utils import DummyObject, requires_backends

class StableDiffusionKDiffusionPipeline(metaclass=DummyObject):
    _backends = ["torch", "transformers", "k_diffusion"]

    def __init__(self, *args, **kwargs):
        requires_backends(self, ["torch", "transformers", "k_diffusion"])
```

(d) Correct code completion of RepoCoder.

Figure 5: Code completion examples of RepoCoder and three major commercial products.

difficult to perform systematic comparison. However, in our case, RepoCoder successfully predicted the correct completion by retrieving a relevant code snippet from the repository context. This demonstrates the need for state-of-the-art tools to effectively leverage repository-level context.