# OpenReview forum: "RepoCoder: Repository-Level Code Completion Through Iterative Retrieval and Generation"
_EMNLP/2023/Conference — EMNLP 2023 Main_

### Official Review · Reviewer_rrM2 · 2023-08-03

**Typos Grammar Style And Presentation Improvements:** I didn't check for grammar issues or …
**Soundness:** 4

**Excitement:**

4: Strong: This paper deepens the understanding of some phenomenon or lowers the barriers to an existing research direction.

**Missing References:**

I didn't check carefully.

**Paper Topic And Main Contributions:**

The paper studies repository-level code generation. The paper presents RepoCoder, an iterative retrieval and generation framework, and introduces a dataset called RepoEval that includes 1600 examples of line and API completion tasks and 373 examples of function completion tasks. While the proposed method is simple and straightforward, its effectiveness cannot be overlooked. The strongest part of the paper is the analysis section which studied different factors to explain the source of improvements.

**Questions For The Authors:**

1. In the API invocation dataset, is all API invocation local to the repository? Or are there global APIs (e.g., torch, numpy) too?
2. Does the length of the sliding window mean the length of the retrieved code chunks (snippets)?
3. What is the maximum number of tokens budgeted for retrieved code snippets in the prompt? It is mentioned that the top 10 snippets are retrieved, but they could be very long.
4. The iterative retrieval and generation approach is not feasible to use practically due to its latency requirements. I want to know the authors' opinions on how we can leverage RepoCoder-like models in real time.

**Reasons To Accept:**

- A simple yet effective approach for repository-level code completion.
- Strong analysis to explain the source of improvements in the code completion tasks.
- Introduced a good dataset in Python for repository-level code completion.

**Reasons To Reject:**

I don't have a reason to reject the work.

**Reproducibility:**

4: Could mostly reproduce the results, but there may be some variation because of sample variance or minor variations in their interpretation of the protocol or method.

**Reviewer Confidence:**

5: Positive that my evaluation is correct. I read the paper very carefully and I am very familiar with related work.

---

> ### Author Rebuttal · Authors · 2023-08-29
>
> We thank you for your kind feedback and for recognizing our work. To address the questions you raised, we have provided detailed responses as follows.
>
> **Q1: In the API invocation dataset, is all API invocation local to the repository? Or are there global APIs (e.g., torch, numpy) too?**
>
> **A:** Yes, all API invocations in our dataset are local to the repository. And there are no other global APIs in the dataset. We ensure the invoked APIs are locally defined within the repository.
>
> **Q2: Does the length of the sliding window mean the length of the retrieved code chunks (snippets)?**
>
> **A:** Yes, the length of the sliding window refers to the size of the retrieved code snippets.
>
> **Q3: What is the maximum number of tokens budgeted for retrieved code snippets in the prompt? It is mentioned that the top 10 snippets are retrieved, but they could be very long.**
>
> **A:** First, the budget for the retrieved code snippets takes half of the prompt's length (mentioned in Section 4.1 Implementation Details). Then, if the top 10 retrieved snippets surpass half of the prompt's length, we only keep the most relevant snippets that fit within the designated length limit (mentioned in Section 2.3 Code Generation).
>
> **Q4: The iterative retrieval and generation approach is not feasible to use practically due to its latency requirements. I want to know the authors' opinions on how we can leverage RepoCoder-like models in real time.**
>
> **A:** RepoCoder indeed introduces concerns about latency. However, RepoCoder's architecture is inherently flexible, allowing for optimizations that make it more suitable for real-time use cases. Here are several avenues:
>
> - Model Quantization or Distillation: Techniques like model quantization or distillation can be employed to reduce the size of the language model while retaining a satisfactory level of performance. Our experiments have indicated that RepoCoder can notably enhance the performance of smaller models, such as CodeGen-350M.
>
> - Hardware Acceleration: Harnessing the power of hardware accelerators, such as GPUs or TPUs, can substantially expedite the inference process of language models, leading to notable reductions in latency.
>
> - Caching and Preprocessing: By pre-caching frequently used code snippets or preprocessing repositories that are frequently accessed, speed can be further improved.
>
> - Adaptive Iterations: Adapting the iteration steps dynamically based on contextual needs and desired latency can yield a balanced compromise between accuracy and speed.

---

### Official Review · Reviewer_ReXb · 2023-08-05

**Soundness:** 4

**Excitement:**

4: Strong: This paper deepens the understanding of some phenomenon or lowers the barriers to an existing research direction.

**Missing References:**

Recently there are some papers on similar topics, such as repo-level benchmark [1], API invocation [2]. The authors could consider discussing them in related work to better tease out research directions.

[1] Liu, Tianyang et al. “RepoBench: Benchmarking Repository-Level Code Auto-Completion Systems.” (2023)

[2] Zhang, Kechi et al. “ToolCoder: Teach Code Generation Models to use API search tools.”(2023)

**Paper Topic And Main Contributions:**

This paper introduces a practical code task: repository-level code completion. Within this context, the authors present a novel benchmark named RepoEval, including scenarios for line, API invocation, and function body completion. This benchmark contributes significantly to the field by providing a platform for in-depth exploration. Additionally, the paper introduces RepoCoder, which employs an iterative retrieval-generation framework to address the complexities of repository-level code completion. The detailed experiments  showcases its effectiveness.

**Questions For The Authors:**

1. Have you evaluated existing code completion commercial applications on your benchmark?

2. Have you evaluated the time efficiency of your method on different settings?

3. This is a concern about potential data leaks: it seems that the training data of GPT-3.5-Turbo is not only up to 2021 as the author stated in the paper, according to [1]. In my opinion, I think the data leak is hard to fix, but it is still essential to claim potential limitations in this paper.

[1] https://openai.com/api-data-privacy (I just noticed that OpenAI has modified the content of this website page and deleted original claims of the training data.)

**Reasons To Accept:**

1. Interesting viewpoint: This paper focuses on the repository-level code completition, which is a research question that fits the actual scene and is worth discussing. The proposed method RepoCoder uses the iterative retrieval-generation framework and achieves good performance.

2. New dataset. I further support the eagerness of the authors to release their dataset to the community. And I believe the proposed RepoEval benchmark will provide a platform for further study.

3. Experiments: The paper provides exhaustive experiments demonstrating the effectiveness of the method.

4. Readability. Moreover, the paper is easy to follow, in my opinion, for a broad NLP community, and it provides visualization of data and the main algorithm.

**Reasons To Reject:**

1. Missing some baselines: For code completion tasks, I suggest that the authors compare with existing code completion commercial applications, such as Copilot. It can be tested on a smaller subset of RepoEval, and it is essential to compare with these state-of-the-art code completion systems.

2. Time efficiency: For code completion tasks, it is also important to focus on the time efficiency. I recommend the authors could add corresponding experiments to make it clear.

3. Missing some related work: Recently there are some papers on similar topics, such as repo-level benchmark [1], API invocation [2]. The authors could consider discussing them in related work to better tease out research directions.

[1] Liu, Tianyang et al. “RepoBench: Benchmarking Repository-Level Code Auto-Completion Systems.” (2023)

[2] Zhang, Kechi et al. “ToolCoder: Teach Code Generation Models to use API search tools.”(2023)

**Reproducibility:**

5: Could easily reproduce the results.

**Reviewer Confidence:**

5: Positive that my evaluation is correct. I read the paper very carefully and I am very familiar with related work.

---

> ### Author Rebuttal · Authors · 2023-08-29
>
> We appreciate your positive feedback and acknowledgment of our efforts. We have taken steps to address the weaknesses and questions you highlighted as follows, aiming to alleviate any concerns you may have.
>
> **W1 & Q1: Missing some baselines: For code completion tasks, I suggest that the authors compare with existing code completion commercial applications, such as Copilot. It can be tested on a smaller subset of RepoEval, and it is essential to compare with these state-of-the-art code completion systems.**
>
> **A:** While we greatly appreciate the suggestion to incorporate comparisons with commercial code completion applications such as Copilot, we wish to underscore the inherent challenges in undertaking such comparisons. Notably, systems like Copilot are sophisticated software solutions that leverage proprietary training data, meticulously tuned lnguage models, and specialized mechanisms to achieve their impressive performance. The inclusion of systematic comparison against such commercial products might introduce biases and unfair comparisons due to the intricate and distinct nature of their underlying complexities.
>
> However, in response to your suggestion, we are open to including comparisons between RepoCoder and other commercial code completion products in the revised version of the paper. By doing so, we hope to extract valuable insights from these comparisons while acknowledging the nuanced context in which they are conducted. We firmly believe that RepoCoder presents a promising avenue for enhancing the capabilities of products like Copilot, offering potential benefits to the broader coding community.
>
> **W2 & Q2: Time efficiency: For code completion tasks, it is also important to focus on the time efficiency. I recommend the authors could add corresponding experiments to make it clear.**
>
> **A:** Undoubtedly, time efficiency is an important demand for real-world applications. While quantifying the time efficiency of RepoCoder presents a challenging task. As outlined in our response to Reviewer rrM2, the time efficiency of RepoCoder is influenced by many engineering strategies such as model quantization for expedited code generation and caching mechanisms for accelerated code retrieval. The implementation of these techniques extends beyond the current scope of our paper. However, we are committed to adding discussions on time efficiency to the Limitations section of our paper.
>
> **W3: Missing some related work: Recently there are some papers on similar topics, such as repo-level benchmark [1], API invocation [2]. The authors could consider discussing them in related work to better tease out research directions.**
>
> **A:** Thank you for bringing this to our attention. We sincerely appreciate your insight regarding the recent papers [1] and [2]. We will definitely incorporate these concurrent works into our related work section to provide a more comprehensive overview of the research landscape.
>
> **Q3: This is a concern about potential data leaks: it seems that the training data of GPT-3.5-Turbo is not only up to 2021 as the author stated in the paper, according to [3]. In my opinion, I think the data leak is hard to fix, but it is still essential to claim potential limitations in this paper.**
>
> Our reference to the training data of GPT-3.5-Turbo being up to 2021 is based on the information provided in the API documentation by OpenAI [4]. We admit that there remains a possibility of data leakage since  the training details of the model are unknown. And we are glad to claim the potential data leakage in the Limitations section.  Importantly, we want to emphasize that despite this concern, the conclusions drawn in our paper retain their validity, as our comparison is fair, and we also incorporates the open-sourced CodeGen model.
>
> **References:**
>
> [1] Liu, Tianyang et al. “RepoBench: Benchmarking Repository-Level Code Auto-Completion Systems.” (2023)
>
> [2] Zhang, Kechi et al. “ToolCoder: Teach Code Generation Models to use API search tools.”(2023)
>
> [3] https://openai.com/api-data-privacy
>
> [4] https://platform.openai.com/docs/models/gpt-3-5

---

### Official Review · Reviewer_LvEZ · 2023-08-10

**Soundness:** 4

**Excitement:**

3: Ambivalent: It has merits (e.g., it reports state-of-the-art results, the idea is nice), but there are key weaknesses (e.g., it describes incremental work), and it can significantly benefit from another round of revision. However, I won't object to accepting it if my co-reviewers champion it.

**Paper Topic And Main Contributions:**

This paper studies the task of repository-level code completion, where both in-file and cross-file context must be utilized. The authors proposed RepoCoder, which incorporates cross-file context through iterative retrieval and generation. Specifically, on top of the standard RAG framework, model-generated code completion from the previous iteration is used as part of the query to improve the subsequent retrieval.

The authors also proposed and constructed a new benchmark dataset for evaluating the task. The dataset consists of line completions, API completions, and function completions with executable unit tests. Results demonstrate more than 10% improvement of RepoCoder over the in-file baseline.

**Questions For The Authors:**

I do not fully understand the criticism of existing approaches in the first introduction paragraph. Can you please elaborate on these points?
1. Why does static analysis have limitations in terms of usability? Static analysis tools are quite reliable and efficient from my perspective.
2. Is there any evidence or reference showing those fine-tuning approaches have generalization problems? If the data can be automatically labeled, e.g. with static analysis, we can potentially leverage the large collection of Github repos to overcome this problem.

**Reasons To Accept:**

1. The proposed RepoCoder outperforms the in-file baseline by a good margin.
2. The evaluation dataset will be useful to the community if open-sourced.
3. The paper is written clearly and easy to follow.

**Reasons To Reject:**

1. The in-file baseline is not strong enough. It would be better to include existing cross-file methods mentioned in the first introduction paragraph.

2. The hyper-parameter tuning in section 4.1 seems performed on the test sets.

3. The proposed method does not fundamentally solve the repo-level code completion problem. As already mentioned in the paper, the method becomes ineffective when exemplars of API calls are missing or misleading, which is quite common in reality. A better approach is probably to resort to API definitions rather than exemplars.

**Reproducibility:**

3: Could reproduce the results with some difficulty. The settings of parameters are underspecified or subjectively determined; the training/evaluation data are not widely available.

**Reviewer Confidence:**

4: Quite sure. I tried to check the important points carefully. It's unlikely, though conceivable, that I missed something that should affect my ratings.

---

> ### Author Rebuttal · Authors · 2023-08-29
>
> We value your feedback and appreciate the differing viewpoints on certain issues you highlighted. To address these concerns, we provide more detailed explanations for each weakness and question. We kindly ask you to reconsider your review in light of these responses.
>
> **W1: The in-file baseline is not strong enough. It would be better to include existing cross-file methods mentioned in the first introduction paragraph.**
>
> **A:** As we mentioned in the Limitations section, "the task of repository-level code completion using language models is relatively new, lacking well-established and reproducible baselines".
>
> We did not include existing cross-file methods mentioned in the Introduction section [1-5], because they can not be easily applied to our scenario. Specially, [1-4] are primarily focused on completing token-level code portions at specific positions. These approaches fall outside the purview of our research's scope. As for the Cocomic method [5], the absence of publicly available data and source code from their work hampers our ability to replicate their methodology.
>
> Given this shortage of suitable baselines, we meticulously designed our experiments to comprehensively evaluate the efficacy of RepoCoder. We introduced the Oracle method for comparative analysis, leveraging its capacity to utilize provided ground truth code completions and retrieve pertinent code from the repository. As a result, it serves as an upper bound performance benchmark for the Retrieval-Augmented Generation (RAG) approach. Our results, as showcased in Table 2 and Table 3, illustrate how RepoCoder's performance, especially with multiple iterations, closely approximates that of the Oracle method. This alignment unequivocally demonstrates RepoCoder's effectiveness.
>
> Additionally, in Section 6.1, we delve deeper into bolstering RepoCoder's efficacy through the implementation of a comparison method called GT-Code. This approach utilizes static analysis tools to directly pinpoint code snippets involving referencing API invocations. The outcomes of this comparison method also demonstrates the good performance of RepoCoder in a more comprehensive context. Hopefully, these explanations will provide a clearer understanding of the sufficiency of our chosen baselines.
>
> **W2: The hyper-parameter tuning in section 4.1 seems performed on the test sets.**
>
> **A:** RepoCoder operates as a framework without any model training. All the hyper-parameters mentioned in section 4.1 are for the inference stage, such as the length of the code window and the sliding size. Given that these parameters are intricately linked to the programming language and contextual scenarios, their adjustment is vital to ensure optimal real-world performance. We will make it clearer in the revised version of our paper.
>
> **W3:  The proposed method does not fundamentally solve the repo-level code completion problem. As already mentioned in the paper, the method becomes ineffective when exemplars of API calls are missing or misleading, which is quite common in reality. A better approach is probably to resort to API definitions rather than exemplars.**
>
> **A:** We respectfully disagree with your assertion. Firstly, it's important to clarify that RepoCoder's code retriever can locate not only API call exemplars but also relevant API definitions from the repository. This capability is illustrated in Figure 2, where RepoCoder successfully identifies the API definition for "run_colmap," enhancing its code completion abilities.
>
> Secondly, though our paper outlines failed cases of RepoCoder when API calls are misleading, these cases do not undermine RepoCoder's fundamental capability to address the repository-level code completion challenge. Instead, they are presented to emphasize the potential for the future improvement of code retrieval.
>
> **Q1:  Why does static analysis have limitations in terms of usability? Static analysis tools are quite reliable and efficient from my perspective.**
>
> **A:** Approaches relying on static analysis [1-4] have limitations in terms of usability, which primarily stem from their constrained code completion contexts. These studies primarily operate within token-level completion scenarios, triggered at specific positions, such as suggesting method names for Java objects. In these specific contexts, static analysis tools demonstrate their reliability and efficiency by parsing potential completion candidates and subsequently ranking. While these methods are not able to perform code completions that involve varying lengths at arbitrary positions.
>
> **Q2: Is there any evidence or reference showing those fine-tuning approaches have generalization problems? If the data can be automatically labeled, e.g. with static analysis, we can potentially leverage the large collection of Github repos to overcome this problem.**
>
> **A:** The concern of the potential generalization problems with fine-tuning approaches is indeed valid. We can elaborate this concern through the cited studies[2, 5, 6] in the Introduction section of our paper.
>
> To begin with, Svyatkovskiy et al. [6] trained a large Transformer model on an extensive corpus of open-source code. However, the model cannot predict unseen variable names. Furthermore, Hellendoorn and Devanbu [2] proposed a dual-model methodology, involving a global model trained on public data and a local model fine-tuned to the specific codebase of the target repository. Despite this approach, the need to retrain the local model for every new repository introduction remains a notable drawback, inhibiting seamless generalization.
>
> Thirdly, Ding et al. [5] presented an approach that imparts repository-level knowledge to the model and employs static analysis tools for data labeling. Nevertheless, their training strategy assumes the presence of project context during code completions, leading to a decline in performance when confronted with scenarios lacking cross-file context, as highlighted in their outlined limitations.
>
> **References:**
>
> [1] Veselin Raychev, Martin Vechev, and Eran Yahav. 2014. Code completion with statistical language models. In Proceedings of the 35th ACM SIGPLAN conference on programming language design and implementation, pages 419–428.
>
> [2] Vincent J Hellendoorn and Premkumar Devanbu. 2017. Are deep neural networks the best choice for modeling source code? In Proceedings of the 2017 11th Joint Meeting on Foundations of Software Engineering, pages 763–773.
>
> [3] Alexey Svyatkovskiy, Ying Zhao, Shengyu Fu, and Neel Sundaresan. 2019. Pythia: Ai-assisted code completion system. In Proceedings of the 25th ACM SIGKDD international conference on knowledge discovery \& data mining, pages 2727–2735.
>
> [4] Alexey Svyatkovskiy, Sebastian Lee, Anna Hadjitofi, Maik Riechert, Juliana Vicente Franco, and Miltiadis Allamanis. 2021. Fast and memory-efficient neural code completion. In 2021 IEEE/ACM 18th International Conference on Mining Software Repositories (MSR), pages 329–340. IEEE.
>
> [5] Yangruibo Ding, Zijian Wang, Wasi Uddin Ahmad, Murali Krishna Ramanathan, Ramesh Nallapati, Parminder Bhatia, Dan Roth, and Bing Xiang. 2022. Cocomic: Code completion by jointly modeling in-file and cross-file context. arXiv preprint arXiv:2212.10007.
>
> [6] Alexey Svyatkovskiy, Shao Kun Deng, Shengyu Fu, and Neel Sundaresan. 2020. Intellicode compose: Code generation using transformer. In Proceedings of the 28th ACM Joint Meeting on European Software Engineering Conference and Symposium on the Foundations of Software Engineering, pages 1433–1443.

---

### Meta-Review · Area_Chair_Df11 · 2023-09-26

**Recommendation:** 4

**Metareview:**

This paper addresses the task of repository-level code completion. They introduce a new dataset – RepoEval, and a new framework RepoCoder that incorporates a similarity-based retriever and a pre-trained code language model.

The paper is clear and well-written. The experiments are thorough. The proposed RepoCoder approach outperforms the in-file baseline. The work is solid and should be of interest to researchers working on code completion.

The question and concern regarding tuning the hyper-parameters on the test data should be addressed in the paper. It should be clarified whether the test data is used for finetuning, and even if it’s only done at the inference stage, it’s still problematic. Adding  a comparison with commercial applications as baselines would strengthen the paper.

---

### Decision · Program_Chairs · 2023-10-07

**Decision:**

Accept-Main

**Comment:**

This paper addresses the task of repository-level code completion. They introduce a new dataset – RepoEval, and a new framework RepoCoder that incorporates a similarity-based retriever and a pre-trained code language model.

The paper is clear and well-written. The experiments are thorough. The proposed RepoCoder approach outperforms the in-file baseline. The work is solid and should be of interest to researchers working on code completion.

The question and concern regarding tuning the hyper-parameters on the test data should be addressed in the paper. It should be clarified whether the test data is used for finetuning, and even if it’s only done at the inference stage, it’s still problematic. Adding  a comparison with commercial applications as baselines would strengthen the paper.